# Molecular Identification of the G-Protein Genes and Their Expression Profiles in Response to Nitrogen Deprivation in *Brassica napus*

**DOI:** 10.3390/ijms23158151

**Published:** 2022-07-24

**Authors:** Yuyu Xie, Yunyou Nan, Ayub Atif, Wencong Hu, Yanfeng Zhang, Hui Tian, Yajun Gao

**Affiliations:** 1College of Natural Resources and Environment, Northwest A&F University, Yangling, Xianyang 712100, China; xieyuyu@nwafu.edu.cn (Y.X.); nanyunyou@nwafu.edu.cn (Y.N.); atifayub.gcuf@gmail.com (A.A.); huwencong2011@163.com (W.H.); 2Hybrid Rapeseed Research Center of Shaanxi Province, Yangling, Xianyang 712100, China; zhangyfcl@163.com; 3Key Laboratory of Plant Nutrition and the Agri-Environment in Northwest China, Ministry of Agriculture, Yangling, Xianyang 712100, China

**Keywords:** G-protein, *Brassica napus*, evolution, nitrogen deficiency

## Abstract

Heterotrimeric guanine nucleotide binding protein (G-protein) consisting of Gα, Gβ, and Gγ subunits is one of the key signal transducers in plants. Recent studies indicated that G-protein has been proposed as an important mediator of nitrogen responses in rice, wheat, and *Arabidopsis*. However, little is known about these G-proteins in *Brassica napus* (*B. napus*), except for three identified G-proteins, BnGA1, BnGB1, and BnGG2. Therefore, the aim of the present study is to characterize the members of the G-protein gene family in allotetraploid *B. napus* and to analyze their expression profiles in response to nitrogen deprivation. In total, 21 G-protein family members were identified in *B. napus*, encoding two Gα, six Gβ, and 13 Gγ. Sequence and phylogenetic analyses showed that although genome-wide triploid events increased the number of genes encoding Gα, Gβ, and Gγ subunits, the gene structure and protein properties of the genes encoding each G-protein subunit were extremely conserved. Collinearity analysis showed that most G-protein genes in *B. napus* had syntenic relationships with G-protein members of *Arabidopsis*, *Brassica rape* (*B. rapa*), and *Brassica oleracea* (*B. oleracea*). Expression profile analysis indicated that Gα and C-type Gγ genes (except *BnGG10* and *BnGG12* were highly expressed in flower and ovule) were barely expressed in most organs, whereas most Gβ and A-type Gγ genes tended to be highly expressed in most organs. G-protein genes also showed various expression patterns in response to nitrogen-deficient conditions. Under nitrogen deficiency, Gα and five C-type Gγ genes were upregulated initially in roots, while in leaves, Gα was downregulated initially and five C-type Gγ genes were highly expressed in different times. These results provide a complex genetic dissection of G-protein genes in *B. napus*, and insight into the biological functions of G-protein genes in response to nitrogen deficiency.

## 1. Introduction

Heterotrimeric guanine nucleotide binding protein (hereafter G-protein) is an evolutionary conserved signal transduction intermediate, composed of three subunits of Gα, Gβ, and Gγ. When receiving external stimulus signals, GDP is replaced by GTP on an associated Gα subunit. At the same time, Gβγ separates from Gα in the form of dimers, and transmits the signal to downstream effectors [1]. After that, the internal GTPase activity in Gα starts to activate Gα-GTP to Gα-GDP, while Gβγ reconnects with Gα to form the inactive G-protein heterotrimer complex [2]. Both Gα and Gβγ can transduce signals by interacting with various intracellular effectors, modulating diverse biological and physiological processes in the plant [3].

In general, the Gα subunit contains a Ras-like domain and a helical domain. The Gβ subunit contains a coiled-coil domain in the N-terminal region and has seven WD40 repeats. The Gγ proteins are subdivided into three types, and the A-type is defined by a CaaX motif. Whereas B-type is similar to A-type but lacks the CaaX motif, C-types contain enriched cysteine residue at the C-terminal and N-terminal similar to the canonical Gγ subunits [4,5]. Although evolution of the G-protein is conserved, there are fewer G-proteins in plants compared with the animal counterparts. While humans contain 23 Gα, 5 Gβ, and 12 Gγ proteins [6], the *Arabidopsis* genome is composed of one Gα, one Gβ, and three Gγ proteins [5]. In addition, in most plants, the number of G-proteins is different. For example, rice has one Gα, one Gβ, and five Gγ [7], while *Brassica rape* (*B. rape*) has one Gα, three Gβ, and five Gγ [8]. Similarly, the *Camelina sativa* genome is composed of three Gα, three Gβ, and nine Gγ [9]. These differences in subunits may lead to the various signaling mechanisms in plants.

Plant G-proteins play fundamental roles in almost every developmental process [10]. Early studies revealed that G-protein acts in controlling hypocotyl elongation, hook angle, rosette diameter, leaf shape [11], auxin-mediated cell division, lateral root proliferation [12], plant height [13], and seed size and number [14,15]. In recent years, plant G-proteins have also been found to be involved in plant responses to nitrogen (N). In wheat, *TaNBP1* is identified to encode Gβ proteins that increase N uptake in response to nitrogen deprivation [16]. Similarly, in rice, the Gγ subunit encoding gene of rice, the *dep1-1* allele increases the absorption and assimilation of nitrogen, and increases the harvest index and grain yield under moderate nitrogen fertilizer levels [17,18,19]. These studies have shown that wheat and rice G-proteins participate in the regulation of nitrogen absorption, transport, and assimilation, increasing nitrogen efficiency, and can be used as a potential biotechnology target for improving nitrogen efficiency. Unlike rice and wheat, the current understanding of the genetic basis of *Arabidopsis* G-protein in nitrogen efficiency remains at the level that it affects nitrogen-related gene expression. For example, *gpa1-5* upregulates the expression of the nitrogen transport gene *AtNRT2.1* [20] and participates in the response to nitrogen by regulating the configuration of roots under different nitrogen concentrations through interaction with the *Atagb1* [21]. In addition, overexpression of *AtAGG3* increases the expression of genes related to nitrogen uptake and transport [22]. However, the functions of G-protein genes in *Brassica napus* (*B. napus*) in mediating plant adaptation to the N-starvation stress are still largely unknown and need to be further investigated.

*B. napus* is an important oilseed crop. Nitrogen is a nutrient massively essential for plant growth, development, and production. For a long time, large amounts of N fertilizers have been applied to arable land around the world to achieve high yields of *B. napus* [23]. However, most of the N applied to the soil is lost because of nitrate leaching, ammonia volatilization, runoff, and denitrification, which has harmful effects on the environment [24,25]. Identification of genes involved in nitrogen signaling in *B. napus* and elucidation of gene expression patterns under low nitrogen conditions could provide new insights into the molecular mechanisms of plant tolerance to low nitrogen stress, as well as provide a theoretical basis for generating crop cultivars with improved nitrogen use efficiency. In this study, 21 G-protein genes in *B. napus* were identified and analyzed for molecular characterization, expression patterns, and expression patterns under low nitrogen conditions. Our results showed that all selected Gα genes and five C-type Gγ genes were extremely sensitive to low nitrogen stress.

## 2. Results

### 2.1. Identification and Analysis of G-Protein Genes in B. napus

Using the five G-protein protein sequences in *Arabidopsis thaliana* (*A. thaliana*) as queries, 21 G-protein genes were identified from *B. napus,* including two Gα subunit genes, six Gβ subunit genes, and 13 Gγ subunit genes (Table 1). The nomenclature used for *B. napus* G-protein genes was based on three known G-protein genes (*BnGA1, BnGB1,* and *BnGG2*—therein, we renamed *BnGG2* as *BnGG8,* but *BnGB1* was not retrieved in our result, so we did not select it in this article) [26,27,28] and the corresponding *A. thaliana G-protein* orthologs. Both BnGα full-length protein sequences were 383 amino acids, with a predicted molecular weight of 44.5 kda and an estimated isoelectric point of 5.78 (Table 1). Amino acid sequence alignments of BnGα with Gα proteins from *A. thaliana* and Rice showed 97.13–96.61% and 75.20–75.47% identity, respectively. Sequence comparison further confirmed the presence of five GTP-binding and GTP-hydrolyzing consensus sequences (G1–G5) and a myristoylated/palmitoylated lipid modification pattern in the N-terminal region, which is required for their plasma membrane localization. In addition, the conserved glutamine (Q) important for the GTPase activity of Gα proteins, the putative ADP ribosylation target (R) for cholera toxin, residue (G) for Gα and the RGS-box interaction were also present in both BnGα proteins (Appendix A).

The BnGβ proteins were encoded by 327–378 amino acids, with a predicted molecular weight of 35.42–41.03 kda and an estimated isoelectric point of 6.66–7.12 (Table 1). The BnGβ proteins were highly conserved, with 90.21–94.69% and 73.56–76.25% identity to the *Arabidopsis* AtAGB1 and rice OsRGB1 proteins, respectively. The coiled-coil hydrophobic structure necessary for its N-terminal interaction with the Gγ subunit were conserved in all BnGβ proteins (Temple and Jones, 2007). Besides, the seven WD repeat motifs were also found to be present in all BnGβ proteins, except for in BnGB3, where the seventh WD was missing (Appendix A).

Thirteen coding sequences of Gγ proteins in *B. napus* were highly divergent in size, ranging from 89–265 amino acids, with a predicted molecular weight of 9.95–28.96 kda and an estimated isoelectric point of 4.23–9.35 (Table 1). Amino acid sequence alignment of BnGγ proteins with *Arabidopsis* (AtAGG1, AtAGG2, and AtAGG3) and rice (OsRGG1, OsRGG2, OsGS3, OsDEP1, and OsGGC2) showed 26.32–94.00% identity with *Arabidopsis* Gγ proteins and 23.64–57.14% of sequence identity with rice Gγ proteins. Sequence comparison further showed that BnGG1, BnGG2, BnGG3, and BnGG4 were the closest homologs of AtAGG1 and OsRGG1; BnGG5, BnGG6, BnGG7, and BnGG8 were homologous to AtAGG2 and OsRGG2; and BnGG9, BnGG10, BnGG11, BnGG12, and BnGG13 were the homologs of AtAGG3, OsGS3, OsDEP1, and OsGGC2, respectively. Sequence alignment further revealed that BnGG1–BnGG8 contained the signature CAAX motif at its C-terminal end, which belonged to A-type Gγ proteins together with *Arabidopsis* AGG1, AGG2, and rice RGG1, whereas BnGG9–BnGG13 belonged to C-type Gγ proteins because of its large size and a very large number of cysteine residues in its C-terminal region. In addition, all BnGγ proteins included a DPLL motif with an important hydrophobic contact with the Gβ subunit protein and an N-terminal helix that forms a coiled-coil structure with Gβ, both of which are conserved in these plant species (Appendix A). Similarly, we also identified 13 candidate G-protein homologs in *Brassica oleracea* (*B. oleracea*) from NCBI by the same method (Appendix A).

### 2.2. Gene Structure Analyses

To compare gene structures of G-protein between *A. thaliana* and *B. napus*, structures of introns/exons were determined by the alignment of genomic DNA with full-length cDNA of *A. thaliana* G-protein and *B. napus* G-protein. The similar gene structures were not only observed in the same G-protein subunits in *B. napus*, but also were found within G-protein homologs in *B. napus* and *A. thaliana* (Figure 1A–C). Among the *B. napus* G-protein genes, all BnGα genes contained 13 exons and 12 introns, and most BnGβ members contained 6 exons and 5 introns, except for *BnGB2* (4 exons and 3 introns each). All A-type Gγ genes contained 4 exons and 3 introns, while members in C-type Gγ genes contained 5 exons and 4 introns, except for *BnGG11* (6 exons and 5 introns).

### 2.3. Phylogenetic Analysis of G-Protein

To analyze the evolutionary relationships of G-protein genes in *Brassica napus*, *Brassica rape*, *Brassica oleracea*, *Brassica nigra* (*B. nigra*), *Arabidopsis*, rice, and *Camelina sativa*, an unrooted phylogenetic tree was constructed using full length amino acid sequences. In total, Ga proteins with publicly reported G-protein sequences from other plant species showed that BnGA1 and BnGA2 proteins were evolutionarily conserved with *BniGα1* (Figure 2A). Upon phylogenetic analysis, all six BnGβ subunits also shared the same clade with the hybridization of two progenitor species, *Brassica rapa* and *Brassica oleracea* [29] (Figure 2B). The relative position of the six BnGβ subunits in the phylogenetic tree further suggested that the six divergent copies of Gβ subunits could be a consequence of a whole genome triplication event [30]. Gγ proteins was divided into two groups, with the A-type Gγ proteins (BnGG1–BnGG8) and C-type Gγ proteins (BnGG9–BnGG13). Moreover, BnGG9–BnGG13 proteins were grouped together with *Arabidopsis* AtAGG3 and rice Gγ protein DEP1, which were recently reported to be yield- and nitrogen-related (Figure 2C) [14,17,18,19,22]; therefore, we conducted subcellular localization to further explore the function of C-type G-protein. First, *BnGG10*, *BnGG11*, and *BnGG12* were selected randomly, transient expression of fused with GFP was performed in *tobacco* (Figure 3), and these proteins were located in cell membranes, which was consistent with the *Arabidopsis* [31]. In a word, a total of 21 G-proteins were identified in *Brassica napus*, with a possibility of 156 Gαβγ combinations, which made it a highly diverse G-protein signaling network, while the model plants *Arabidopsis* and rice have only three and five such combinations [4].

### 2.4. Chromosomal Location and Gene Duplication of G-Protein

The 21 G-protein genes were distributed on 11 *Brassica napus* chromosomes randomly (Figure 4A), with 12 genes in the A subgenome and 9 genes in the C subgenome. Three G-protein genes each were identified on chromosomes A3 and C3, and two were found on A1, A4, A9, C1, C7, and C8. Only one G-protein gene was found on chromosomes A2, A8, and A10. In addition, the synteny relationship of G-protein genes was analyzed using the genome information from *B. napus*, *A. thaliana*, *B. rapa*, and *B. oleracea* (Figure 4B). A total of 12 pairs of syntenic paralogs and 78 pairs of syntenic orthologs were found in these genomes. Eight, one, and three pairs of G-protein syntenic paralogs were obtained in *B. napus*, *B. rapa*, and *B. oleracea*, respectively, and none in *A. thaliana*. Ten pairs of G-protein syntenic orthologs were observed between *B. napus* and *A. thaliana*, 23 pairs of G-protein syntenic orthologs were observed between *B. napus* and *B. rapa*, and 27 between *B. napus* and *B. oleracea*. In addition, 11 pairs of G-protein syntenic orthologous genes were observed between *B. rapa* and *B. oleracea*, suggesting possible chromosomal rearrangement and reduction of the G-protein gene code after a whole genome triplication event.

Of 21 identified *B. napus* G-protein duplicated pairs, none of the genes were confirmed to be tandem duplicated genes. Additionally, nine genes (*BnGB2–BnGB5* and *BnGG1*, *BnGG4*, *BnGG9*, *BnGG12*, and *BnGG13*) were segmentally duplicated genes (Appendix A). Interestingly, all segmental duplicated genes belonged to Gβ and Gγ, which accounted for 42.9% of the G-protein genes in *B. napus*. Based on the above results, it could be inferred that segmental duplication played a predominant role in the expansion of the *B. napus* G-protein family.

To better understand the evolutionary constraints acting on the G-protein gene family, the nonsynonymous rate (Ka), synonymous rate (Ks), and evolutionary constraint (Ka/Ks) ratio were estimated for *B. napus* (Figure 5) [32]. Our results showed that the Ka/Ks ratio for Gα and Gγ subunit genes were <1, indicating that these genes are functionally conserved and had experienced strong purifying selective pressure during evolution, while the Ka/Ks ratio for the homologous genes in the Gβ subunit genes was >1, which might have been subjected to positive pressures after the whole genome triplication (WGT) event in *B. napus*, making BnGβ subunit genes’ functions more diverse. In addition, although the Ka/Ks ratio of Gγ subunit genes was <1, the Ka/Ks ratio of A-type Gγ subunit genes was two or three times larger than that of C-type Gγ subunit genes, indicating that A-type Gγ subunit genes had evolved more diverse functions.

### 2.5. Analysis of Cis-Acting Element in G-Protein Genes’ Promoters

Cis-elements are short, specific DNA sequences that bind with TFs to regulate the transcription of genes. Cis-elements identification is a key step in understanding transcriptional regulation mechanisms. To further study the potential regulatory mechanisms of *B. napus* G-protein genes, we screened for cis-elements in the 2000 bp upstream promoter sequences of these genes using PlantCARE [33]. Forty-seven types of cis-elements were detected in the promoters of G-protein genes. Except for essential cis-elements, the others (light response elements, five hormone responsive elements, four abiotic stress response elements, MBSI, O2-site, circadian, MSA-like, CAT-box, and GCN4_motif) were analyzed (Figure 6) and 42 physiological-process-related cis-elements were observed in most of the G-protein promoters. All G-proteins possessed at least one light-responsive element. Gα subunit members might be associated with stress responsiveness, especially *BnGA1*, which had one or more stress-response elements. Gβγ as a dimer might have a potential response under abscisic acid, MeJA, and anaerobic responsiveness. In addition, *BnGB2*, *BnGB4*, and *BnGG3* contained more hormone-responsive elements, while *BnGB5*, *BnGB6*, and *BnGG12* contained more stress-responsive elements compared with other Gβ and Gγ subunit members. In any event, the cis-element analysis illustrated that *B. napus* G-protein genes could respond to different physiological processes involving hormone regulation, stress responsiveness, and aging.

### 2.6. Expression Patterns of G-Protein Genes in Different Tissues

Using the RNA-seq data, a heatmap of 21 G-protein genes, represented in different tissues and organs in *B. napus*, was established by TBtools (Figure 7). Calculation of the expression levels of the G-protein genes revealed diverse expression patterns in different subunits. All Gα subunit genes (*BnGA1* and *BnGA2*), were barely expressed in any tissues or organs. Three Gβ subunit genes including *BnGB1*, *BnGB4*, and *BnGB5* were highly expressed in all of the tissues, while *BnGB2* and *BnGB6* showed high expression levels only in some tissues, and *BnGB3* was barely expressed in any tissue. All A-type Gγ subunit genes, except for *BnGG2*, *BnGG4*, and *BnGG7*, showed relatively high expression levels in all organs. Most C-type Gγ subunit genes (*BnGG9*, *BnGG11*, and *BnGG13*) were barely expressed in any tissues or organs. However, *BnGG10* and *BnGG12* were highly expressed in vegetative organs, such as flowers and ovules, which were grouped together with *AGG3*, *GS3*, and *DEP1* genes related to the regulation of grain yield in the phylogenetic tree (Figure 2C). In general, members of Gβ and A-type Gγ subunits often shared different expression patterns, whereas Gα and C-type Gγ genes were relatively conserved, which was consistent with their Ka/Ks ratio.

### 2.7. Expression Patterns of G-Protein Genes under Nitrogen Deficiency

To investigate the expression patterns of *B. napus* G-protein genes under nitrogen deficiency, seedling leaves and seedling roots were harvested at various time points (0 to 7 d) after culturing in a Hoagland nutrient solution with high nitrogen (normal N condition, CK) and low nitrogen (N deficient condition, LN) content. In leaves (Figure 8), under LN treatment, Gα subunit gene *BnGA1* was significantly downregulated immediately and subsequently showed no significant difference with CK after 24 h. *BnGG10* was significantly downregulated first and then significantly upregulated after 24 h. However, the other members of other C-type Gγ subunit genes were significantly upregulated first, and subsequently *BnGG9* showed a downregulated expression and *BnGG11* showed no significant difference with CK, while *BnGG12* and *BnGG13* remained upregulated.

However, the responses of most G-protein genes in roots to LN treatment were highly different to those in leaves (Figure 8 and Figure 9). Under LN treatment, *BnGA1* was more sensitive to nitrogen deficiency under 3 h and 24 h treatment in roots than in leaves, which was upregulated at 3 h, downregulated at 24 h, and had no significant difference with CK in other time points (Figure 8 and Figure 9). Among five C-type Gγ subunit members, all were upregulated first and then three (*BnGG9*, *BnGG10*, and *BnGG11*) showed no significant difference with CK, while *BnGG12* and *BnGG13* were downregulated after LN treatment (Figure 9).

## 3. Discussion

G-protein, as a key signal transducer, is an evolutionarily conserved signaling intermediate that regulates signal recognition and transduction, hormone perception, and stress responses [10]. Studies on plants’ G-proteins have improved the knowledge of essential physiological and agronomic properties. With the availabilities of the whole genome sequences of many plants, several G-protein families have been identified, such as five, seven, nine, and eight G-protein genes in *Arabidopsis*, rice, *B. rape*, and *B. nigra*, respectively [5,8,34,35]. However, little is known about the G-protein family in *B. napus* except for three known genes (*BnGA1, BnGB1,* and *BnGG2*) that have been reported [26,27,28]. The current study identified 21 *B. napus* G-protein family genes and analyzed their structure, chromosomal location, phylogeny, gene duplication, Ka/Ks ratio, cis-elements, and expression patterns in different tissues and nitrogen deficiency. The study provides comprehensive information on the *B. napus* G-protein gene family and will aid in understanding the functional divergence of G-protein genes in *B. napus*.

*B. napus* has a polyploid genome formed by interspecific crosses between *B. rapa* and *B. oleracea* [36], *B. rapa* has nine G-protein genes (one Gα, three Gβ, and five Gγ) [9], and *B. oleracea* has thirteen G-protein genes (one Gα, five Gβ, and seven Gγ) (Appendix A), thus *B. napus* is expected to have multiple G-protein members. Brassica lineage have undergone a whole-genome triploidy (WGT) event with many genomic arrangements and gene losses, which greatly shaped the genomic framework of existing Brassica species [37]. Although, the WGT event shaped the Gβ genes (six homologs) and Gγ genes (thirteen homologs) in *B. napus*, only two Gα genes (*BnGA1* and *BnGA2*) were identified in the present study. Interestingly, as found in *B. napus*, there was only one typical Gα subunit gene in both the *B. rapa* (Bra007761) and *B. oleracea* (LOC106307271) progenitors.

Gene structure plays a crucial role in the evolution of several gene families. In the present study, BnGα proteins might have maintained functional conservation, with highly similar gene structures, conserved domains, and close phylogenetic relationships between the two Gα subunit genes. Similarly, six BnGβ proteins and thirteen BnGγ proteins retained the necessary structural domains (WD40 and GGL, respectively), but the sequences of some BnGβ and BnGγ genes had been separated from each other. For example, *BnGB2* contained fewer exons than the remaining BnGβ genes; *BnGG11* had one more exon than the other C-type Gγ. Therefore, functional divergence might have occurred in these duplicated genes, including neo- and sub-functionalization. Accordingly, these were also confirmed by different Ka/Ks ratios of each subunit genes; Gα had the most conservative evolution and Gβ had the most positive evolution. Interestingly, although Gγ had evolved conservatively, A-type Gγ had evolved faster than C-type Gγ. Thus, all these results suggested that the functions of Gβ and A-type Gγ might be diversified, and Gα and C-type Gγ proteins might have retained functional conservation.

The expression patterns of G-protein genes in different tissues have been investigated in many species, such as *Arabidopsis*, rice, *B. rapa*, and *B. nigra* [8,34,38]. In *B. rapa* and *B. nigra*, Gα genes (*BraA.Gα1* and *BniB.Gα1*) express ubiquitously and maintain an almost similar expression pattern across different developing stages [8,34]. In rice, *OsRGA1* are highly expressed in leaves of seedlings, representing the most abundant *OsRGA1* transcripts [38]. *BjuA.Gα1* in *B. juncea* are highly expressed in leaves, roots, male catkins, xylem, seeds, and female catkins [15]. Gβγ have ubiquitous, overlapping but distinct expression profiles across plant development [34]. Thus, there is no uniform gene expression pattern for plant G-protein genes. According to the RNA-seq data of *B. napus* (Figure 6), although the two canonical BnGα genes retained an almost low expression profile across different tissue types, the multiple members of the BnGβ and BnGγ subunit genes exhibited an incongruous spatio-temporal expression pattern in various tissues, indicating that different BnGβγ heterodimers might have diverse functions across plant growth and development. In the current study, *BnGB1* and *BnGB5* were highly and indiscriminately expressed in all organs, such as flower, ovule, pistil, silique. Similar expression levels with several A-type Gγ genes (*BnGG1*, *BnGG5*, *BnGG3*, *BnGG6*, and *BnGG8*) were also found in *B. napus*. Interestingly, in recent years, the C-type Gγ subunit genes have been shown to regulate organ development, seed size and shape, and oil production in plants [13,14,17,18,19,22,39,40]. Similar to previous reports [8,34], in this study, C-type Gγ subunit genes (*BnGG10* and *BnGG12*) showed specific expression activity in flower, ovule, and pistil, the main source organ during the podding and flowering stage in *B. napus*, suggesting that *BnGG10* and *BnGG12* genes might be involved in seed development.

G-protein genes involved in the response to nitrogen have been reported in several plants [16,17,18,19,20,21]. In *Arabidopsis*, GPA1 and GCR1 cooperatively regulate the nitrate response [20]. In rice, DEP1, a Gγ subunit-encoded protein, can efficiently balance carbon and nitrogen metabolism and thus increase tiller and seed number while improving nitrogen use efficiency [17,18,19]. In wheat, *TaNBP1*, a gene encoding a Gβ subunit, is adapting to nitrogen starvation by regulating nitrogen acquisition [16]. Thus, we selected some genes with high homology within previous studies. Because the genes with high homology of wheat Gβ subunit gene (*TaNBP1*) in *B. napus* displayed annotation without the key word “guanine nucleotide-binding subunit” and *BnGA2* had no specific primer, we chose *BnGA1* and five C-type Gγ genes with high homology. In this study, we found that all G-protein genes we chose were sensitive to N starvation, either highly expressed or suppressed in leaves and roots. Interestingly, in roots, all G-protein were upregulated in the initial period under low nitrogen (Figure 9). These upregulated G-protein genes might play a key role in nitrogen uptake and utilization in response to the short time demand for nitrogen in *B. napus* during low nitrogen. However, *BnGA1* was suppressed in leaves, providing insights that *BnGA1* might be improving nitrogen absorption in roots during low nitrogen. Of these C-type Gγ subunit genes, *BnGG9* and *BnGG11* were highly expressed in the first 3 h, and *BnGG10* and *BnGG12* were highly expressed in 3 d and 7 d, only the *BnGG13* were highly expressed in leaves during 3 h–7 d (Figure 8). These suggested that C-type Gγ subunit genes, especially *BnGG13*, might have been involved in coordinating the N balance of leaves during various periods, for storing nitrogen to coordinate leaf growth and photosynthesis, or for increasing nitrogen partitioning to promote plant growth and improve yield. The different expression profiles of these G-protein genes in roots and leaves after nitrogen-deficiency treatment suggested that these C-type Gγ subunit members had experienced sub-functionalization during the polyploidy evolution of *B. napus*, pointing to different nitrogen utilization mechanisms among *B. napus* G-proteins.

Compared with the expression pattern represented by RNA-seq data, the expression profile generated by qRT-PCR was not completely equal to that. The difference of expression pattern might be caused by the fact that these genes were involved in stress responses. Similar to a previous study [8], most C-type BnGγ are the barely abundant transcripts among Gγ genes across all the tissue types tested, whereas it was preferentially upregulated during nitrogen-stress conditions.

In summary, a genome-wide analysis of the *B. napus* G-protein family was performed and 21 *B. napus* G-protein genes were identified. Analyses of G-protein genes on gene structures, protein character, phylogeny, chromosomal location, synteny, cis-elements, and expression patterns in different organs and nitrogen stress were conducted to define their biological functions. We found that G-protein genes had various expression patterns, and all Gα and five C-type Gγ genes were extremely sensitive to nitrogen deficiency. This provides insights into the biological functions of *B. napus* G-protein genes in response to nitrogen deficiency.

## 4. Materials and Methods

### 4.1. Identification of the G-Protein Family Members in Brassica napus Genome

The whole *Brassica napus* and *Brassica oleracea* protein sequence was downloaded from NCBI (https://www.ncbi.nlm.nih.gov/genome/?term=Brassica+napus, accessed on 26 July 2020). To identify *Brassica napus* G-protein candidates, a BLASTP-algorithm-based [41] search using *Arabidopsis* G-protein amino acid sequences as queries was conducted with an e-value ≤ 1 × 10^−20^. To avoid missing probable G-protein members, the hidden Markov model (HMM) analysis [42] was used for the search. We downloaded the HMM profiles of Gα (PF00503), Gβ (PF00400), and Gγ (PF00631) from the Pfam protein family database (http://pfam.xfam.org/, accessed on 26 July 2020) and used it as the query (*p* < 0.001) to search the *B. napus* protein sequence data. Additionally, keywords “guanine nucleotide-binding protein” were employed to search against NCBI. After removing all of the redundant sequences, the output putative G-protein protein sequences were submitted to Pfam (http://www.ebi.ac.uk/Tools/pfa/pfamscan/, accessed on 26 July 2020) and SMART (http://smart.embl-heidelberg.de/, accessed on 26 July 2020) to confirm the conserved Gα (G-alpha), Gβ (WD40), and Gγ (GGL) domain. All of the non-redundant and high-confidence genes were assigned as *B. napus* G-proteins. These G-protein genes were named on the basis of known *B. napus* G-protein genes (*BnGA1*, *BnGB1*, and *BnGG2*) [26,27,28] and the corresponding *Arabidopsis* G-protein orthologs.

The same methods were applied to identify the candidate G-proteins in *B. oleracea* genomes in NCBI.

### 4.2. Sequence Analysis and Structural Characterization

All the *B. napus* G-protein sequences were submitted to ExPASy (http://web.expasy.org/protparam/, accessed on 26 July 2020) to predict the number of amino acids, molecular weights, and theoretical isoelectric points (pI).

TBtools was used to count the map of the exon/intron structures [43].

### 4.3. Phylogenetic Analysis and Subcellular Localization of G-Protein

The full-length amino acid sequences from various plant species including *Brassica rape* [8], *Brassica nigra* [34], *Arabidopsis* [5], *Oryza sativa* [5], *Camelina sativa* [9], *Brassica oleracea*, and *Brassica napus* were identified as derived from published paper and current study, respectively. Multiple sequence alignments of the G-protein sequences were performed using the ClustalW tool in MEGA 7.0 [44]. Based on these alignments, three phylogenetic trees, comprising 10 Gα, 22 Gβ, and 46 Gγ protein full-length sequences, respectively, were constructed using the neighbor-joining method, Poisson correction, pairwise deletion, and bootstrapping (1000 replicates; random seeds) as the required parameters.

The coding sequence of *BnGG10*, *BnGG11*, and *BnGG12* was amplified from the cDNA of *Brassica napus* cultivar “Zhongshuang11 (ZS11)” and cloned into a green fluorescent protein (GFP) fusion expression vector pEGFP via an enzyme (SpeI) digestion-ligation method. All the primers were synthesized by TSINKE Biotech and are listed in Appendix A. The Agrobacterium tumefaciens strain GV3101 harboring the 35S:BnGG10, 11, and 12-GFP constructs were injected into tobacco leaves, and cultured at 28 °C for two days. The florescence images were obtained by using a confocal microscope (Olympus, FV3000, Tokyo, Japan).

### 4.4. Chromosomal Localization and Gene Duplication

Chromosomal locations of the *B. napus* G-protein genes were obtained from NCBI using the accession number. The MapChart2.2 (https://mapchart.net/, accessed on 1 July 2022) was then used to draw chromosomal location diagrams for these genes [45].

A multiple collinearity scan toolkit (MCScanX) was adopted to analyze the gene duplication events with the default parameters. The synteny relationship of the G-protein in *B. napus*, *A. thaliana*, *B. rape*, and *B. oleracea* were constructed using TBtools [43].

### 4.5. Calculation of the Ka/Ks Values of G-Protein Genes

The CDS sequences of orthologous G-protein gene pairs between *B. napus* and *A. thaliana* were first aligned using ClustalW. Then, the nonsynonymous rate (Ka), synonymous rate (Ks), and evolutionary constraint (Ka/Ks) between the orthologous gene pairs were calculated according to their CDS sequence alignments by using Dnasp5 in the Ka/Ks_calculator program [46].

### 4.6. Analysis of Cis-Acting Element in B. napus G-Protein Genes’ Promoters

The upstream sequences (2 kb) of the *B. napus* G-protein genes were retrieved from the NCBI and then submitted to PlantCARE (http://bioinformatics.psb.ugent.be/webtools/plantcare/html/, accessed on 26 July 2020) to analyze the cis-acting element [33].

### 4.7. Plant Materials and Nitrogen Deprivation Treatments

Seeds of *Brassica napus* cultivar ZS11 were germinated in plant growth chambers and transplanted into pots in full-strength Hoagland solution (7.5 mmol/L) in a plant growth chamber with a thermo-photoperiod of 25 °C for 16 h/18 °C for 8 h (light/dark) [47]. One-month-old seedlings were transferred into new full-strength Hoagland solution (7.5 mmol/L, CK) and modified N-deficient Hoagland nutrient solution (0.19 mmol/L, LN) for treatment [48]. All nutrition solutions were renewed every three and a half days. Three biological replicates of leaves and roots were harvested after 0, 3, 12, 24 h, 3, and 7 d of treatment. All samples were immediately frozen in liquid nitrogen and stored at −80 °C.

### 4.8. RNA Isolation, RNA-Sequencing, and Quantitative Reverse-Transcription PCR

Total RNA was extracted from the organ using an RNA extraction kit (Bioteke, Beijing, China), and cDNA was synthesized from 1 µg of total RNA using M-MLV transcriptase (Bioteke, Beijing, China) according to the manufacturer’s instructions.

The *B. napus* ‘ZS11’ RNA-seq data (BioProject ID PRJNA394926) were downloaded from the NCBI-SRA database (https://www.ncbi.nlm.nih.gov/sra/, accessed on 26 July 2020) to study the organ-specific expression patterns of *B. napus* G-protein genes. The RNA-seq data (Appendix A) included different tissues. To render the data suitable for cluster displays, absolute fragments per kilobase of exon per million reads mapped (FPKM) values were divided by the mean of all of the values, and the ratios were used to construct the heatmaps by TBtools.

Reverse transcription quantitative PCR (RT-qPCR) was performed on the QuantStudio 5 Real Time PCR System (ABI Quantstudio 5, Foster city, CA, USA). Gene-specific primer sequences for the G-protein genes were obtained from NCBI Primer-blast [49] (Appendix A). Each reaction was carried out in biological triplicate in a reaction volume of 10 µL containing 1.0 µL of gene-specific primers (1.0 µM), 2.0 µL of cDNA, 5 µL of SYBR green, 0.1 µL of ROX Reference Dye II, and 3 µL of sterile distilled water. The PCR program was as follows: 95 °C for 10 min; 40 cycles of 10 s at 95 °C and 30 s at 60 °C; and then melt curve 65 °C to 95 °C, increment 0.5 °C for 5 s. Melting curves were generated to estimate the specificity of these reactions (Appendix A). Relative expression levels were calculated using the 2^−∆∆Ct^ method, with *BnACTIN7* used as an internal control due to its stable expression in leaf and root samples of *B. napus* under nitrogen deficiency [50].

### 4.9. Statistical Analysis

All experiments were repeated at least three times independently and data were analyzed using the Office 2010 software. Statistical analyses were calculated with SPSS25 software using Student’s *t*-test with a *p*-value threshold of 0.05 (^★^) or 0.01 (^★★^).

## Figures and Tables

**Figure 1 ijms-23-08151-f001:**
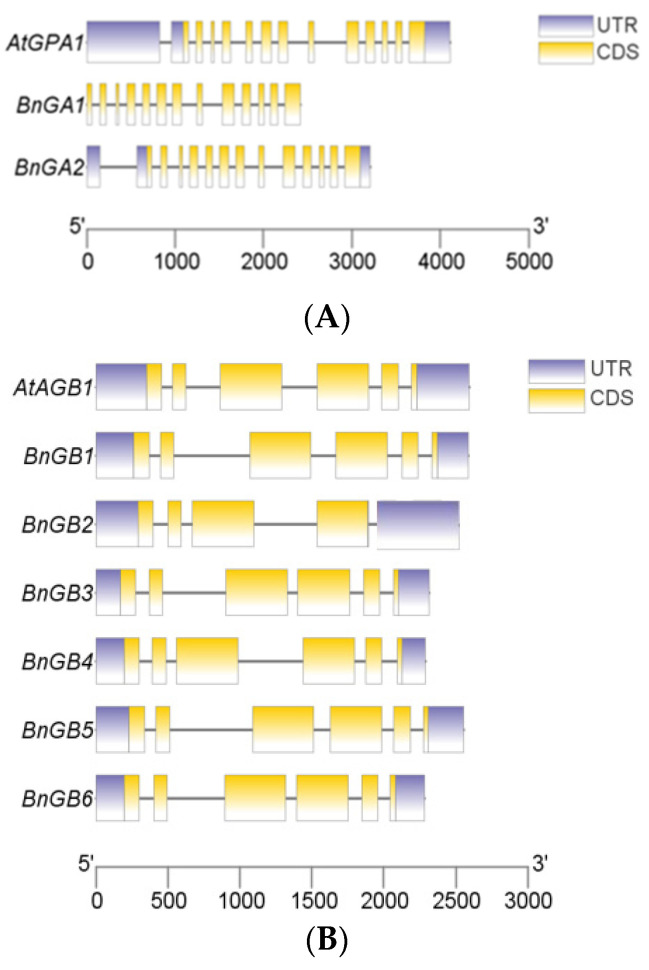
Gene structure analysis of Gα (**A**), Gβ (**B**), and Gγ (**C**) genes between *A. thaliana* and *B. napus*. Yellow boxes represent exons and black lines represent introns. The upstream/downstream region of G-protein genes are indicated in purple boxes. The length of exons can be inferred by the scale at the bottom.

**Figure 2 ijms-23-08151-f002:**
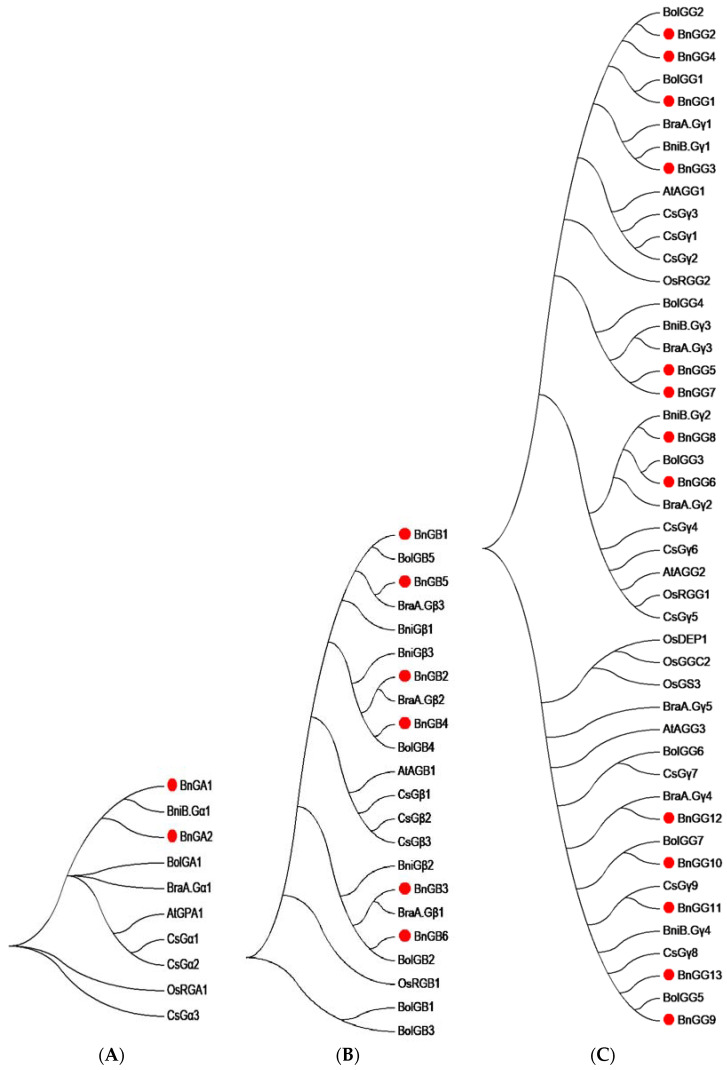
Evolutionary relationship of *B. napus* G-protein. The phylogenetic analysis of (**A**) Gα, (**B**) Gβ, and (**C**) Gγ proteins isolated from *B. napus*, with their corresponding G-protein subunit proteins in *B. rape* (*Bra*), *B. oleracea* (*Bol*), *B. nigra* (*Bni*), *Arabidopsis* (*At*), *Oryza sativa* (*Os*) and *Camelina sativa* (*Cs*), was performed using the neighbor-joining method in MEGA 7.0, with 1000 bootstrap replications. The red dots represent *B. napus* G-protein.

**Figure 3 ijms-23-08151-f003:**
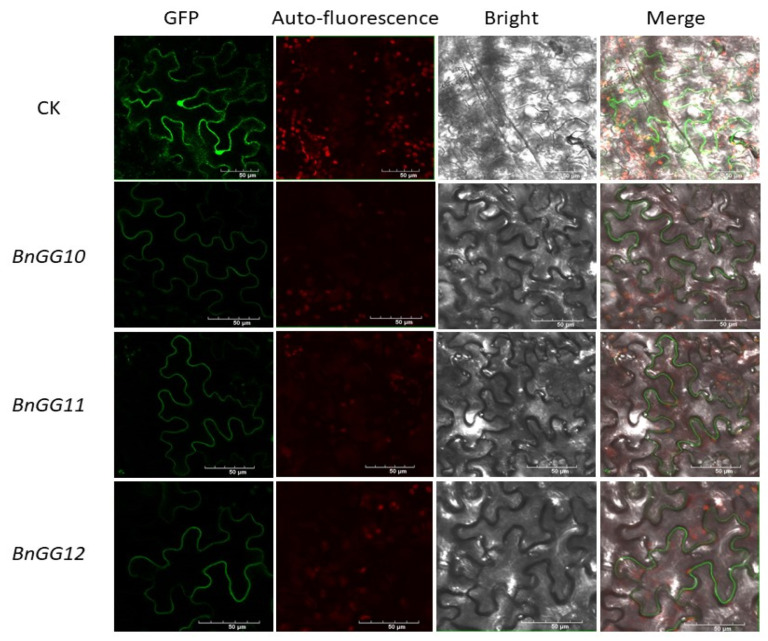
Subcellular localization of three selected *B. napus* G-protein genes in tobacco cells.

**Figure 4 ijms-23-08151-f004:**
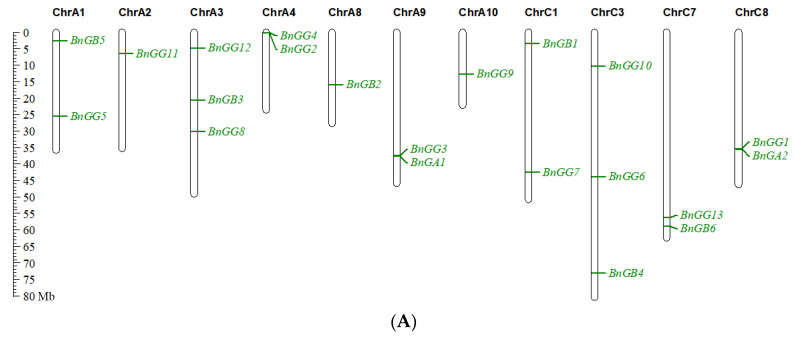
Chromosomal location of *B. napus* G-proteins (**A**) and syntenic relationships of G-protein in *A. thaliana*, *B. napus*, *B. rapa*, and *B. oleracea* (**B**). (**B**): blue lines represent Gα synteny, orange lines represent Gβ synteny, green lines represent Gγ synteny.

**Figure 5 ijms-23-08151-f005:**
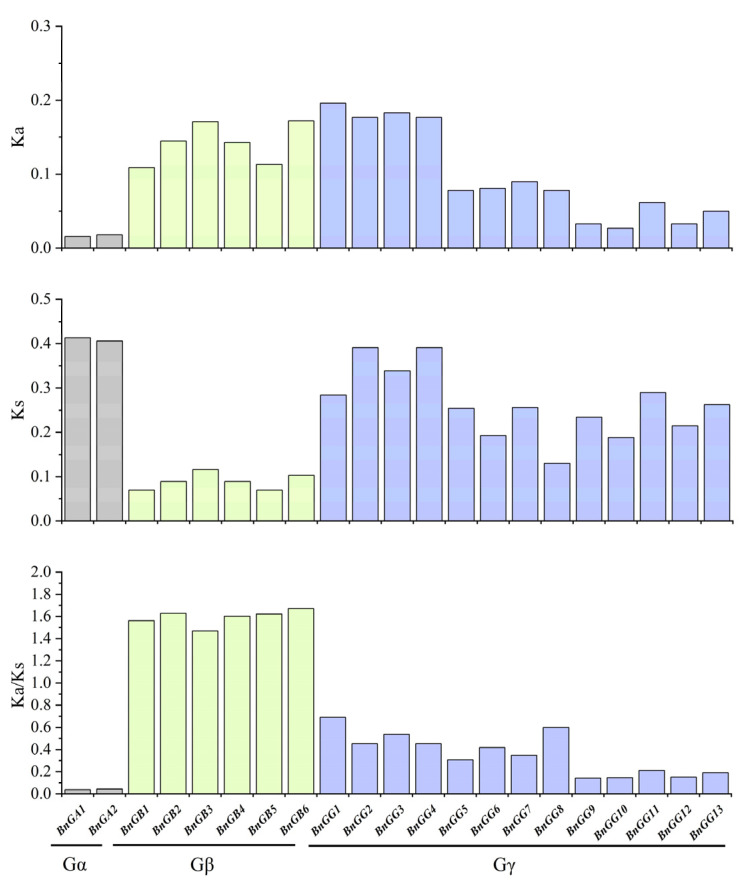
Nonsynonymous rate (Ka), synonymous rate (Ks), and evolutionary constraint (Ka/Ks) ratio between *A. thaliana* G-proteins and the corresponding orthologs in *B. napus*. Gray, green and purple represents Gα subunit genes, Gβ subunit genes, and Gγ subunit genes, respectively.

**Figure 6 ijms-23-08151-f006:**
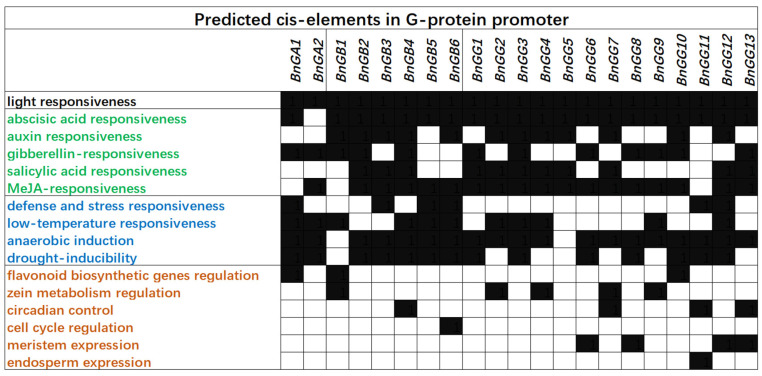
Predicted cis-elements in *B. napus* G-protein promoters. Promoter sequences (2000 bp) of 21 G-proteins are analyzed by PlantCARE. Cis-elements were identified in the promoters of all 21 G-proteins and were classified into four main groups (black, green, blue, and orange, respectively, represent groups). Black means that gene has one cis-element at least, and white means that there is no cis-element.

**Figure 7 ijms-23-08151-f007:**
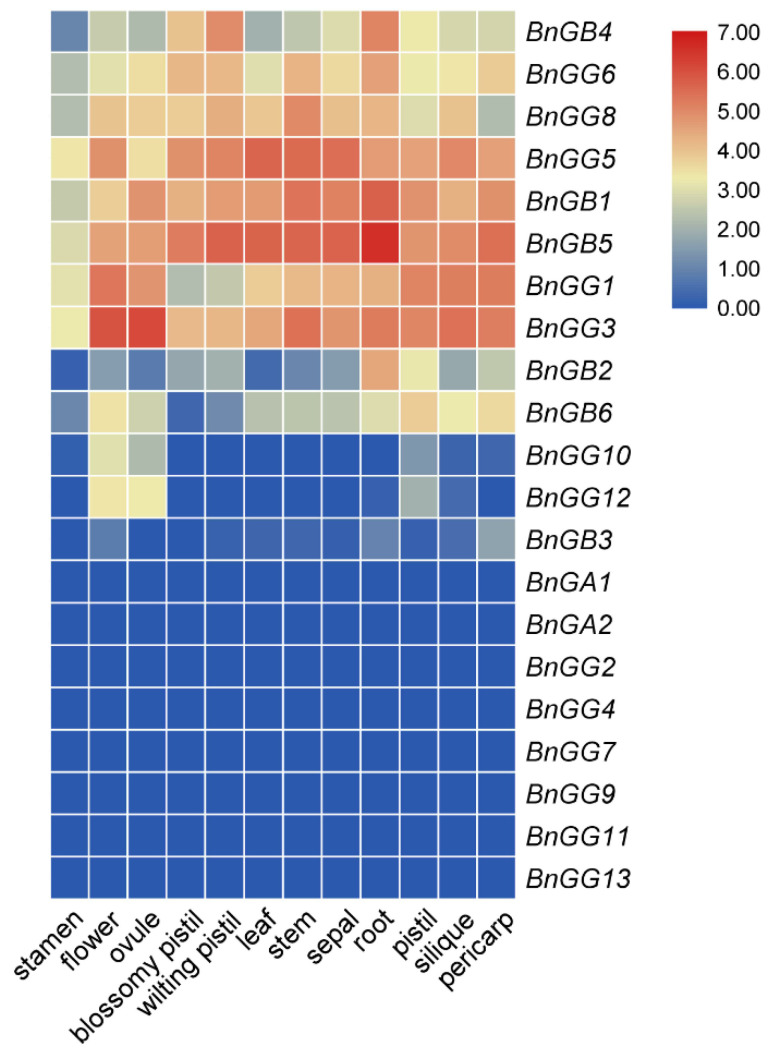
Expression profiles of G-proteins in different tissues and organs of *B. napus*. FPKM values of G-protein genes were transformed by log2 and the heatmap was constructed by TBtools.

**Figure 8 ijms-23-08151-f008:**
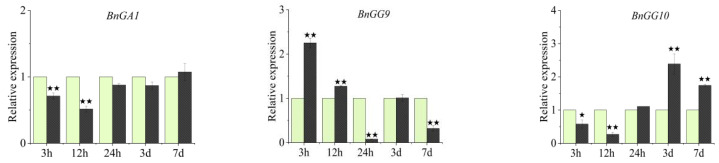
Expression profiles of G-protein genes in leaves under normal and nitrogen-deficient conditions. Quantitative RT-PCR was used to investigate the expression levels of one Gα gene (*BnGA1*) and five C-type Gγ genes (*BnGG9*, *BnGG10*, *BnGG11*, *BnGG12*, *BnGG13*). Leaves were collected under normal-growth and nitrogen-deficient conditions at 0 h, 3 h, 12 h, 24 h, 3 d, and 7 d after treatments. Data were normalized to the *BnActin7* gene and vertical bars indicate standard deviation. Compared with the expression of G-protein genes in CK condition at each time point, ^★^ represents the significant level, ^★^^★^ represents the extremely significant level.

**Figure 9 ijms-23-08151-f009:**
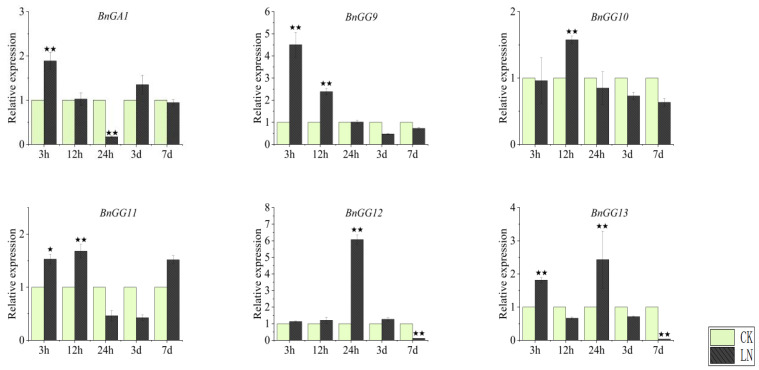
Expression profiles of G-protein genes in roots under normal and nitrogen-deficient conditions. Quantitative RT-PCR was used to investigate the expression levels of one Gα gene (*BnGA1*) and five C-type Gγ genes (*BnGG9*, *BnGG10*, *BnGG11*, *BnGG12*, and *BnGG13*). Roots were collected under normal-growth and nitrogen-deficient conditions at 0 h, 3 h, 12 h, 24 h, 3 d, and 7 d after treatments. Data were normalized to the *BnActin7* gene and vertical bars indicate standard deviation. Compared with the expression of G-protein genes in CK condition at each time point, ^★^ represents the significant level, ^★^^★^ represents the extremely significant level.

**Table 1 ijms-23-08151-t001:** Features of G-protein genes identified in *B. napus*.

Name	Gene ID	Chr.	Genomic Location	AA	MW (kDa)	pI
*BnGA1*	106368534	A9	37616342–37619503	383	44.51	5.83
*BnGA2*	106416165	C8	35538778–35541981	383	44.52	5.83
*BnGB1*	106355793	C1	3466087–3468674	377	40.95	7.12
*BnGB2*	106361031	A8	15871124–15873524	327	35.42	6.81
*BnGB3*	106393802	A3	20605662–20607978	378	41.03	6.84
*BnGB4*	106434467	C3	73099901–73102187	378	40.98	6.66
*BnGB5*	111200422	A1	2483391–2485944	377	40.96	7.12
*BnGB6*	111207516	C7	58909449–58911731	378	40.95	6.66
*BnGG1*	106415923	C8	35416454–35417485	89	10.03	4.32
*BnGG2*	106444922	A4	160931–162114	93	10.44	4.87
*BnGG3*	111200375	A9	37488644–37489890	89	9.95	4.23
*BnGG4*	106444926	A4	148539–149721	102	11.55	8.96
*BnGG5*	106378744	A1	25494898–25496358	101	11.46	5.67
*BnGG6*	106416661	C3	43845900–43847294	101	11.22	4.69
*BnGG7*	106453742	C1	42473926–42475664	156	17.81	9.35
*BnGG8*	106444151	A3	30215792–30217228	101	11.16	4.69
*BnGG9*	106370440	A10	12691034–12694126	243	26.59	8.71
*BnGG10*	106385330	C3	10289041–10292301	236	25.67	8.71
*BnGG11*	106425695	A2	6534486–6537500	265	28.96	8.78
*BnGG12*	106443216	A3	4798356–4803485	237	25.76	8.71
*BnGG13*	106409403	C7	56275611–56278720	246	26.86	8.63

## Data Availability

The RNA-Seq data used in this study are available in the Sequence Read Archive (SRA) at NCBI (SRA accession: PRJNA394926) repository.

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
