# Peer review of "Molecular Identification of the G-Protein Genes and Their Expression Profiles in Response to Nitrogen Deprivation in *Brassica napus"

_ijms, 2022, doi:10.3390/ijms23158151_

Round 1

Reviewer 1 Report

In this publication by Xie et al., the author identified G proteins involved in GPCR signaling pathways in Brassica napus. The authors hypothesized that G proteins are important mediators for nitrogen deprivation. The authors share data that shows changes in G protein levels inside the plants with nitrogen deprivation. But the results observed do not display any pattern or correlation in time-course profiles (increase or decrease). 

Major Comments

A) In the data shown by qRT PCR, the increase and decrease in G protein levels looks random, are the observations noise or real effect? Can authors show a correlation with other housekeeping genes? Housekeeping genes like beta-actin do change expression under stressful conditions. 

B) Can the author describe why the data from RNA seq doesn't correlate with qRT PCR? How was the analysis performed on RNA seq data? 

C) Abbreviations need to be defined at the first use like Ka, Ks, etc.

Minor comments

A) In Figure 2A, remove framing around the figure. Also in the Legend, the gene needs to be capitalized.

B) Line 190 - typo for C-type

C) Figure 5: The scale lacks a size description on the scale bar. There is an arrow on the figure.

Author Response

Response to Reviewer 1 Comments

Dear reviewer,

We are very grateful to you for reviewing the manuscript so carefully. We have considered your suggestions and made some changes.

Point1: In the data shown by qRT PCR, the increase and decrease in G protein levels looks random, are the observations noise or real effect? Can authors show a correlation with other housekeeping genes? Housekeeping genes like beta-actin do change expression under stressful conditions. 

Response 1: Thank you for your suggestion and questions.

We also considered a lot about this question and referred to some papers and found that in qRT PCR data under adversity conditions, some articles used 0 h as a control, e.g.,

①Yang J, Zhou J, Zhou H J, et al. Global survey and expressions of the phosphate transporter gene families in Brassica napus and their roles in phosphorus response[J]. International journal of molecular sciences, 2020, 21(5): 1752.

②Miao L, Gao Y, Zhao K, et al. Comparative analysis of basic helix-loop-helix gene families among Brassica oleracea, Brassica rapa, and Brassica napus[J]. BMC Genomics, 2020, 21(1): 1-18.

③Lu S, Fadlalla T, Tang S, et al. Genome-wide analysis of phospholipase D gene family and profiling of phospholipids under abiotic stresses in Brassica napus[J]. Plant and Cell Physiology, 2019, 60(7): 1556-1566.

And some use normal conditions as a control, e.g.,

①Liu M, Chang W, Fan Y, et al. Genome-wide identification and characterization of NODULE-INCEPTION-like protein (NLP) family genes in Brassica napus[J]. International journal of molecular sciences, 2018, 19(8): 2270.

②Wan Y, Wang Z, Xia J, et al. Genome-wide analysis of phosphorus transporter genes in Brassica and their roles in heavy metal stress tolerance[J]. International journal of molecular sciences, 2020, 21(6): 2209.

③Tong J, Walk T C, Han P, et al. Genome-wide identification and analysis of high-affinity nitrate transporter 2 (NRT2) family genes in rapeseed (Brassica napus L.) and their responses to various stresses[J]. BMC plant biology, 2020, 20(1): 1-16.

Considering that the low nitrogen treatment in this paper had a longer time range from 0 h to 7 d, we believed that the expression of G protein genes under normal nitrogen conditions might change with time, so it is not clear whether the change of G protein expression under nitrogen stress is from the change of time or caused by nitrogen stress, so it might be inaccurate to use 0 h as the control, and we chose to use normal nitrogen conditions as the control to improve the accuracy of the experiment, so we consider our qPCR data as the real effect.

There are different optimal housekeeping genes under different stresses, for example, the best ranked reference genes are PP2A and TIP41 for salt stress, TIP41 and ACT7 for heavy metal (Cr6+) stress, PP2A and UBC21 for drought stress, F-box and ZNF for salicylic acid stress, TIP41, ACT7 and PP2A for methyl jasmonate stress, and TIP41 and ACT7 for abscisic acid stress (Yang H, Liu J, Huang S, et al. 2014;), but whether there are more suitable housekeeping genes under nitrogen stress, there are few studies on this aspect. However, according to previous studies, actin is the traditional housekeeping gene, showing small expression changes in different tissues, stresses and hormone treatments as well as in Brassica napus (B. napus) cultivars (Yang H, Liu J, Huang S, et al. 2014). Furthermore, in most of the paper listed above (Lu S, Fadlalla T, Tang S, et al.2019; Yang J, Zhou J, Zhou H J, et al.2020; Liu M, Chang W, Fan Y, et al.2018; Wan Y, Wang Z, Xia J, et al.2020; Tong J, Walk T C, Han P, et al.2020), actin7 is chosen to be used as a housekeeping gene. And based on our qPCR results, we also found that the expression of actin7 in roots and leaves changed very little with time under different nitrogen concentration treatments. Therefore, we believed that it was feasible to select actin as a housekeeping gene.

Point 2: Can the author describe why the data from RNA seq doesn't correlate with qRT PCR? How was the analysis performed on RNA seq data? 

Response 2: Thank you for your interesting questions.

We guessed that there could be two reasons. On the one hand, the difference in expression patterns could be caused by the fact that these genes are involved in the stress response. Similar to previous studies (Arya G C, Kumar R and Bisht N C, 2014), most of the C-type BnGγ is the least abundant transcript of the Gγ gene in all tissue types tested, while it is preferentially upregulated under stress conditions. This point had been discussed in the penultimate paragraph of our discussion. On the other hand, the RNA seq data are from NCBI, and we are not sure in which period of the data of root and leaf in B. napus were sampled, and it is possible that B. napus G protein genes are expressed at low levels in that period of sampling.

The B. napus ‘ZhongShuang11′ RNA-seq data (BioProject ID PRJNA394926) were downloaded from NCBI-SRA database (https://www.ncbi.nlm.nih.gov/sra/) and then the B. napus genome in NCBI was used as a reference for sequence comparison, data filtering, quality control, and finally expression quantification and normalization to obtain the FPKM values of the target genes in different B. napus tissues.

Point 3: Abbreviations need to be defined at the first use like Ka, Ks, etc.

Response 3: Thank you for your suggestion, we have defined the abbreviations at the first use.

Point 4: In Figure 2A, remove framing around the figure. Also in the Legend, the gene needs to be capitalized.

Response 4: Thanks for the suggestion, we have removed framing around the Figure 2A and have capitalized the genes in the legend.

Point 5:Line 190 - typo for C-type

Response 5: Thank you for your suggestion, we have corrected the spelling of Line 190 - typo to C-type.

Point 6:Figure 5: The scale lacks a size description on the scale bar. There is an arrow on the figure.

Response 6: Thank you for the suggestion. We have added a size description on the scale bar and removed the arrows from the figure.

Reviewer 2 Report

The authors identified 21 G-proteins in Brassica napus and got some interesting results. The method seems fine. The results present well.

Comments:

1. Why not use the commonly used B. napus genome for analysis directly, e.g., Darmor or ZS11? These genome annotations are more valuable and their gene IDs are more friendly.

2. The Ka/Ks results are interesting, all GB's Ka/Ks>1. Strongly recommend using a figure to point out these genes' CDS alignment.

No comments on other parts.

Author Response

Responds to the reviewer 2 comments

Point 1: Why not use the commonly used B. napus genome for analysis directly, e.g., Darmor or ZS11? These genome annotations are more valuable and their gene IDs are more friendly.

Response 1: Thank you for your suggestion and question, and we agree with what you said.

On the one hand, the NCBI database is one of the commonly used plant genome databases, and the Brassica napus (B. napus) genome in NCBI uses the genome of B. napus variety “ZS11”, which is also currently used in several papers, e.g.,

Nan Y, Xie Y, Atif A, et al. Identification and Expression Analysis of SLAC/SLAH Gene Family in Brassica napus L[J]. International Journal of Molecular Sciences, 2021, 22(9): 4671.

Xue Y, Jiang J, Yang X, et al. Genome-wide mining and comparative analysis of fatty acid elongase gene family in Brassica napus and its progenitors[J]. Gene, 2020, 747: 144674.

He Y H, Zhang Z R, Xu Y P, et al. Genome-Wide Identification of Rapid Alkalinization Factor Family in Brassica napus and Functional Analysis of BnRALF10 in Immunity to Sclerotinia sclerotiorum[J]. Frontiers in plant science, 2022, 13: 877404-877404.

On the other hand, in result 2.6 we used B. napus “ZS11” transcriptome data from the NCBI database when analyzing tissue expression. To be consistent with this, our B. napus genome was finally selected from the NCBI database for the B. napus “ZS11” genome.

Point 2. The Ka/Ks results are interesting, all GB's Ka/Ks>1. Strongly recommend using a figure to point out these genes' CDS alignment.

Response 2: Thank you for the suggestion, we have used a figure to point out these genes' CDS alignment.

Reviewer 3 Report

The study was focused on molecular identification of the G-protein genes and their expression profiles in response to nitrogen deprivation in Brassica napus. The Authors revealed that sequence and phylogenetic analyses showed that although genome-wide triploid events increased the number of genes encoding Gα, Gβ, and Gγ subunits, the gene structure and protein properties of the genes encoding each G-protein subunit are extremely con-served. Collinearity analysis indicated that most G-protein genes in B. napus had syntenic relationships with G-protein members of Arabidopsis, B. rapa, and B. oleracea.

The paper is quite interesting, however, I recommend the following minor improvements:

-        The Introduction is too long, it should be presented in more concise form,

-     -   In Materials and Methods, a subsection describing the statistical tests used should be added,

-        - In the Supplementary file, melting curves after real-time qRT-PCR analyses, should be included (e.g. TIFF files),

-        - Figure 1 and Table 1 may be removed into the Supplementary file,

-        - Moderate English style and grammar changes are required.

Author Response

Responds to the reviewer 3 comments

Point 1: The Introduction is too long, it should be presented in more concise form,

Response 1: Thank you for your suggestion. We have reorganized the introduction to make it clear and concise in the manuscript.

Point 2:  In Materials and Methods, a subsection describing the statistical tests used should be added,

Response 2: Thank you for your suggestion. We've added statistical analysis to the Materials and Methods.

Point 3:  In the Supplementary file, melting curves after real-time qRT-PCR analyses, should be included (e.g., TIFF files),

Response 3: Thank you for the suggestion. The results of the melting curves have been added to Figure S2.

Point 4: Figure 1 and Table 1 may be removed into the Supplementary file

Response 4: Thank you for your suggestion. We have removed Figure 1 into the Supplementary file. However, considering that Table 1 is important in the full text, we have left Table 1 in the main text.

Point 5: Moderate English style and grammar changes are required.

Response 5: Thank you for your suggestion. We have modified to the English style and grammar of the manuscript.